# Can Computer-Aided Design and Computer-Aided Manufacturing Integrating with/without Biomechanical Simulation Improve the Effectiveness of Spinal Braces on Adolescent Idiopathic Scoliosis?

**DOI:** 10.3390/children10060927

**Published:** 2023-05-24

**Authors:** Qian Zheng, Chen He, Yan Huang, Tao Xu, Yi Jie, Christina Zong-Hao Ma

**Affiliations:** 1Department of Rehabilitation Medicine, Tongji Hospital, Tongji Medical College, Huazhong University of Science and Technology, Jiefang Avenue, Wuhan 430030, China; qianzhengtongji@163.com (Q.Z.); rehabcc@163.com (T.X.); 2Institute of Rehabilitation Engineering and Technology, University of Shanghai for Science and Technology, Shanghai 200093, China; hechen@usst.edu.cn; 3Department of Biomedical Engineering, The Hong Kong Polytechnic University, Hong Kong SAR 999077, China; yi620.jie@connect.polyu.hk; 4Department of Rehabilitation Engineering, The Fifth Affiliated Hospital, Zhengzhou University, Zhengzhou 450052, China; 5Research Institute for Smart Aging, The Hong Kong Polytechnic University, Hong Kong SAR 999077, China

**Keywords:** adolescent idiopathic scoliosis (AIS), computer-aided design and computer-aided manufacturing (CAD/CAM), spine, brace, systematic review, meta-analysis

## Abstract

The CAD/CAM technology has been increasingly popular in manufacturing spinal braces for patients with adolescent idiopathic scoliosis (AIS) in clinics. However, whether the CAD/CAM-manufactured braces or the CAD/CAM-manufactured braces integrating with biomechanical simulation could improve the in-brace correction angle of spinal braces in AIS patients, compared to the manually manufactured braces, has remained unclear. The purpose of this systematic review and meta-analysis was to compare the in-brace correction angle of (1) computer-aided design and computer-aided manufacturing (CAD/CAM)-manufactured braces or (2) the CAD/CAM-manufactured braces integrating with biomechanical simulation with that of (3) manually manufactured braces. The Web of Science, OVID, EBSCO, PUBMED, and Cochrane Library databases were searched for relevant studies published up to March 2023. Five randomized controlled trials (RCTs) or randomized controlled crossover trials were included for qualitative synthesis, and four of them were included for meta-analysis. The meta-analysis effect sizes of the in-brace correction angle for CAD/CAM versus manual method, and CAD/CAM integrating with biomechanical simulation versus the manual method in the thoracic curve group and the thoracolumbar/lumbar curve group were 0.6° (mean difference [MD], 95% confidence intervals [CI]: −1.06° to 2.25°), 1.12° (MD, 95% CI: −8.43° to 10.67°), and 3.96° (MD, 95% CI: 1.16° to 6.76°), respectively. This review identified that the braces manufactured by CAD/CAM integrating with biomechanical simulation did not show sufficient advantages over the manually manufactured braces, and the CAD/CAM-manufactured braces may not be considered as more worthwhile than the manually manufactured braces, based on the in-brace correction angle. More high-quality clinical studies that strictly follow the Scoliosis Research Society (SRS) guidelines with long-term follow-ups are still needed to draw more solid conclusions and recommendations for clinical practice in the future.

## 1. Introduction 

Adolescent idiopathic scoliosis (AIS) is a three-dimensional structural spinal deformity in the frontal, sagittal and axial planes that occurs in patients in or around the pubertal growth spurt period [1,2] and the etiopathogenesis of this disorder has so far remained unknown [3,4]. AIS affects approximately 2.4–5.1% of children, and females are more prone to develop progressive curves than males [5,6]. Long-term follow-up studies have indicated that patients with scoliosis may have a higher prevalence of back pain and worsening pulmonary function than individuals without scoliosis if the abnormal curve angle becomes extremely large [7,8].

Brace treatment (or spinal orthoses treatment) and physiotherapeutic scoliosis-specific exercises (PSSE) are currently the widely accepted nonsurgical treatment methods for scoliosis [9,10,11]. Such treatments aim to slow down or stop abnormal scoliosis curves from further progression in adolescents [2,12]. The earliest scoliosis orthoses/braces have been manufactured manually by prosthetists and orthotists working/practicing at hospitals, rehabilitation centers, or clinics [13]. With the increasing maturity of manually manufactured technology, various types of scoliosis braces have emerged, including the Milwaukee brace, Boston brace, Charleston brace, Cheneau brace, and their improved designs [14]. These manually manufactured braces have played an important role in the treatment of scoliosis. However, the manual manufacturing method has the limitation of heavily relying on the clinical experience of the orthotist, prolonged manufacturing time that usually lasted for several days, and difficulty in standardizing the brace design features across different orthotists or patients [13,15,16,17].

In order to address the shortcomings of manual manufacturing methods and improve the treatment effect of scoliosis braces, computer-aided design and computer-aided manufacturing (CAD/CAM) technology have been integrated into the traditional manual manufacturing methods. Previous studies have supported that the CAD/CAM technology has the advantages of simpler measurement procedures, less manufacturing time (approximately 1/3 of the time of the manual method [18]), more comfortable and hygienic experience for patients during the “casting” stage, and more standardized manufacturing processes [17,18]. Due to the strong compatibility of CAD/CAM technology and the improved treatment effect of scoliosis braces, more recently, other advanced technologies that have been integrated with the CAD/CAM technology, such as biomechanical simulation, can help orthotists and physicians to further optimize the brace designing process [19,20,21,22,23]. Specifically, the biomechanical simulation technology added an additional step of simulating the curve correction on the finite element model (FEM) of AIS patient’s trunk, with different corrective forces and areas, which enables several rounds of cast modification until getting the optimal brace correction effect before manufacturing the braces [19,20,21,22,23]. The CAD/CAM technology integrating with/without more advanced technologies have also been developed to improve the in-brace correction of the AIS braces [19,20,22].

However, from the cost-benefit aspect, the CAD/CAM technology with/without more advanced technologies requires the high expenses of purchasing the system and the additional technical training for clinicians [17], which might be a heavy burden for most hospitals, rehabilitation centers, or clinics. This may also be the main reason restricting the promotion of these new technologies at the moment. To facilitate the decision-making of whether it is worthwhile to purchase and apply the CAD/CAM with/without more advanced technologies in hospitals, rehabilitation centers or clinics, it is important and necessary to explore if the CAD/CAM technology integrating with/without more recent technologies could achieve better treatment efficacy than the manual method, or even replace the manual method in manufacturing AIS braces potentially. Nevertheless, to the best knowledge of the authors, so far none of the previous systematic reviews and meta-analyses have specifically searched for the published high-quality clinical trials (i.e., randomized controlled trials, RCTs) and investigated whether braces manufactured by CAD/CAM with/without integrating with biomechanical simulation could have the comparable or even better in-brace correction angle in AIS patients than the manually manufactured ones.

Thus, the objectives of this systematic review and meta-analysis were (1) to search, identify, and assess high-quality studies through a systematic and standardized approach; (2) to qualitatively synthesize and compare the influencing factors of the in-brace correction angle (e.g., obey the Scoliosis Research Society (SRS) inclusion criteria [1]), and the effectiveness of AIS braces manufactured by the CAD/CAM technology integrating with/without biomechanical simulation and the manual method; and (3) to quantitatively analyze and compare the in-brace correction angle of manually manufactured braces AIS braces with the CAD/CAM technology manufactured braces integrating with/without biomechanical simulation. Based on the findings of the previously published high-quality clinical studies, this systematic review and meta-analysis will provide more solid and high-level evidence for future decision-making and clinical practice, and inspire future researches in the field.

## 2. Materials and Methods

### 2.1. Study Design and Registration

This is a systematic review and meta-analysis that conducted following the PRISMA guidelines and registered in the PROSPERO registry for systematic review protocols (reference number: CRD42022306360) [24].

### 2.2. Search Strategy

The following electronic databases were searched for relevant papers that were published up to March 2023: Web of Science, OVID, EBSCO, PUBMED, and Cochrane Library. The Population, Intervention, Comparison, and Outcome (PICO) question asked was “Can the CAD/CAM or the CAD/CAM integrating with biomechanical simulation manufactured braces improve the in-brace correction angles compare to the manually manufactured braces on AIS patients?”. The indexing terms and free-text words included “adolescent idiopathic scoliosis”, “Computer-aided design”, “braces”, “spinal orthoses”, and “randomized controlled trial”. The full list of the used indexing terms and keywords and an example of the searching strategy employed for PubMed is listed in Table 1. The search was limited to English-language articles. The reference lists of the included articles and the searched published systematic reviews or meta-analyses were also searched and reviewed for any missed studies.

### 2.3. Inclusion Criteria

The inclusion criteria of published studies were (1) Study design: randomized controlled trials or randomized controlled crossover trials; (2) Population: participants or patients diagnosed with AIS; (3) Intervention and comparator: investigating the comparison of the treatment effects of the AIS braces manufactured by the CAD/CAM method versus manual method, and the CAD/CAM method with biomechanical simulation versus manual method; and (4) Outcomes: outcome measures included the in-brace correction angle/rate in the coronal plane.

### 2.4. Exclusion Criteria

The exclusion criteria were (1) studies with mixed populations (such as AIS patients with additional diseases that may cause abnormalities in the musculoskeletal system); and (2) studies with other mixed treatments.

### 2.5. Screening of Studies

The bibliographic details of all retrieved articles were stored in an EndNote file, and the duplicate references were firstly removed. Secondly, two reviewers/authors (Q Zheng and Y Huang) read the titles and abstracts independently and removed the studies that did not fulfill the inclusion and exclusion criteria. Thirdly, the two reviewers (Q Zheng and Y Huang) read the full text of the remaining articles to further screen them and included the studies for this review. Disagreements were resolved by Q Zheng and Y Huang’s face-to-face discussion. If there were any disagreement or a consensus could not be reached, a third reviewer was consulted (CZH Ma) to make the final decision [24].

### 2.6. Assessment of Studies

Three reviewers (Q Zheng, Y Huang, and C He) used the Physiotherapy Evidence Database score (PEDro) system to assess the methodological quality and risk of bias of the included articles [25]. The PEDro scale includes 10 items for assessment of trial quality based on whether the trials reported the randomization procedure, concealed allocation, blinding of patients, blinding of assessors, adequate follow-up, intention-to-treat analysis, between-group comparability, between-group statistical comparison, and point estimate and variability or not [26,27]. The PEDro scores of four points or more were classified as “sufficient quality,” whereas studies with scores of three points or less were classified as “insufficient quality” and were subsequently excluded from the meta-analysis in this review. Disagreements were resolved by Q Zheng, Y Huang, and C He face-to-face through discussion and then by consultation with CZH Ma [24].

### 2.7. Qualitative Analysis and Synthesis

Three reviewers (Q Zheng, Y Huang, and C He) independently extracted data from the included studies. The extracted data included author, year of publication, article type, number of participants, interventions, comparators, primary outcomes and follow-up time. Disagreements were resolved by Q Zheng, Y Huang, and C He face-to-face through discussion and then by consultation with CZH Ma [24].

### 2.8. Quantitative Analysis (Meta-Analysis)

The primary outcome measure of the study was the in-brace correction angle, which was calculated as follows: Cobb pre-brace—Cobb in-brace [16,28]. The in-brace correction angles were extracted from both the experimental and control patient groups of the included studies. The Review Manager 5.3.5 (Sun Microsystems Inc., Copenhagen, Denmark, 2014) was used in this review to perform the meta-analysis of the in-brace correction angle for the included studies. The heterogeneity test was performed using the I^2^ statistic. An I^2^ greater than 50% represented substantial heterogeneity, and the random-effect models were used for data analysis [29]. Otherwise, the fixed-effect model was used. The mean difference (MD) and 95% confidence intervals (95% CI) were also calculated.

## 3. Results

As shown in the PRISMA flow diagram [24] (Figure 1), a total of 619 studies were initially identified from the systematic search. Among these studies, 609 studies were excluded, including 139 duplicate references. Then 467 studies were excluded by reading the titles and abstracts of the references. Finally, the full texts of the remaining thirteen studies were reviewed. Based on the inclusion and exclusion criteria, five more studies were further excluded, and the remaining five studies were eligible for the qualitative and quantitative analysis in this review.

### 3.1. Study Characteristics

As shown in Table 2, the included five studies were all RCTs or randomized controlled crossover trials with good methodological quality [21,22,23,30,31].

Among the five included studies with 139 AIS participants, two studies were RCTs [23,30] and three studies were randomized controlled crossover trials [21,22,31] (Table 3). The sample size ranged from 6 [22] to 48 [23] AIS patients. A majority of the AIS participants were females in these studies.

### 3.2. Scoliosis Research Society (SRS) Inclusion Criteria

None of the five included trials described the inclusion criteria of patients with reference to the SRS inclusion criteria (i.e., aged 10 years or older when the brace was prescribed; the Risser sign ranged from 0 to 2 and the primary curve angles ranged from 25° to 40°) [4]. Upon analyzing the information of the five included trials (Table 4), including the participants’ age (12 ± 1 [22,23,30] to 13 ± 2 [21]), the curve angles (24 ± 5 [31] to 36 ± 11 [23]), and the Risser sign (1 [21,22,23,31] to 2 [30]), we identified that the five included trials obeyed the SRS inclusion criteria [21,22,23,30,31]

### 3.3. Influencing Factors for In-Brace Angle

The curve type, curve flexibility, and brace design are the main influencing factors for both the in-brace correction angle and the initial in-brace correction rate (in-brace correction angle/pre-brace angle × 100%) [32]. These influencing factors in the five included studies were as follows: (1) All the five included studies recruited AIS patients with matched thoracic or double curves in the experimental group and control groups; (2) Only Cottalorda et al., 2005 assessed the curve flexibility before the brace treatment with an anteroposterior X-ray assessment in supine and standing positions, while the other four studies did not; and (3) The brace designs in the five included trials were all rigid braces, including underarm customized TLSO braces [31], Boston braces [21,22,23] and Hong Kong braces (modified version of Boston brace) [30] (Table 3).

### 3.4. Qualitatively Synthesize

The average in-brace correction angles of CAD/CAM with/without biomechanical simulation and manually manufactured braces were all greater than the measurement error of approximately 3–5° [33,34] in the five included trials. Upon comparing the in-brace correction angle of braces manufactured by different methods, most results showed larger in-brace correction angles of the CAD/CAM integrating with/without the biomechanical simulation method. However, Cottalorda et al., 2005 [31] reported a larger in-brace correction angle for manually manufactured braces than that of the CAD/CAM manufactured braces. Meanwhile, Blais et al., 2012 [22] and Cobetto et al., 2014 [21] reported a larger in-brace correction angle for thoracic curves of the manually manufactured braces than that of the CAD/CAM braces integrating with biomechanical simulation (Table 4).

### 3.5. Quantitative and Meta-Analysis

The standard deviation of the averaged in-brace correction angle was not reported in the study by Cobetto et al., 2014 (Table 4). Thus, only the remaining four studies [22,23,30,31] were eligible and included for the final meta-analysis of this review. These four studies were further divided into two subgroups (“Manually manufactured braces” vs. “CAD/CAM-manufactured braces” and “Manually manufactured braces” vs. “CAD/CAM combining with biomechanical simulation manufactured braces”), following the recommendation of the “Cochrane Handbook for Systematic Reviews of Interventions” that meta-analysis is the statistical combination of results from two or more separate studies [35].

A total of two trials (totally 70 AIS participants) compared the effectiveness between the manually manufactured braces and the CAD/CAM-manufactured braces [30,31] (Figure 2). The MD (fixed-effect model) of the in-brace correction angle was 0.60° (95% CI: −1.06° to 2.25°, level of heterogeneity I^2^ = 50%).

A total of two trials (totally 53 AIS participants) compared the effectiveness between the manually manufactured braces and the CAD/CAM-manufactured braces integrating with biomechanical simulations [22,23]. Due to the fact that these two articles separately recorded the in-brace correction angle of the thoracic curves and the thoracolumbar/lumbar curves, the results were further divided into two subgroups based on the scoliotic curve level: (1) the thoracic curve group (Figure 3) and (2) the thoracolumbar/lumbar curve group (Figure 4). The MD (random-effect model) of the thoracic curve group in-brace correction angle was 1.12° (95% CI: −8.43° to 10.67°, level of heterogeneity I^2^ = 70%), and that of the thoracolumbar/lumbar curve group in-brace correction angle was 3.96° (95% CI: 1.16° to 6.76°, level of heterogeneity I^2^ = 0%).

## 4. Discussion

This systematic review and meta-analysis compared the in-brace correction angle of AIS braces manufactured by (1) the manual method, (2) the more advanced CAD/CAM method, and (3) the state-of-the-art CAD/CAM method integrating with biomechanical simulation to treat AIS patients upon involving the high-quality RCTs and randomized crossover clinical trials.

### 4.1. Methodological Quality of the Included Studies

The five included studies were either RCTs or randomized crossover clinical trials, which can be considered as studies with high methodological quality. Regarding the study design of the included trials, the five included trials had recruited AIS patients that fulfilled the SRS inclusion criteria [4]; however, the limited number of the sample size may limit the generalization of the conclusions of this systematic review and meta-analysis.

### 4.2. Influencing Factors for In-Brace Angle and Qualitative Analysis

Except for the manufacturing methods, the influencing factors (curve type, curve flexibility, and brace design) influencing the in-brace correction angle shall also be controlled and/or matched among the different subject/participant groups when qualitatively comparing the in-brace correction rate of the braces manufactured by different methods. Currently, three studies were randomized controlled crossover trials [21,22,31], and the patients in two groups had the same curve type, curve flexibility and brace design. However, the curve flexibility assessments were lacking in the other two RCTs [23,30], which may have made the comparison of the in-brace correction angle between different manufacturing methods less objective. Meanwhile, in the randomized controlled crossover trials, the confounding factors, including but not limited to the testing order and the interval time of different manufactured methods, should be avoided as much as possible.

### 4.3. Quantitative and Meta-Analysis

The CAD/CAM-manufactured braces shall not be considered more worthwhile than the manually manufactured brace, based on the in-brace correction angle. Through analyzing the synthesized results of two subgroups, we found that the synthesized mean difference of the in-brace correction angle of CAD/CAM-manufactured braces was 0.6° greater than the manually manufactured braces and is less than the measurement error (i.e., 3–5°) [33,34]. The 95% confidence interval (−1.06° to 2.25°) contained zero. This finding is also in line with a narrative review published in 2019 which concluded that there has been insufficient evidence yet to conclude that CAD/CAM integrating with/without biomechanical simulation methods provides significantly better clinical outcomes than those of the conventional methods in the treatment of scoliosis curves. Specifically, in this meta-analysis, for the thoracolumbar/lumbar curve group, the CAD/CAM-manufactured braces integrating with biomechanical simulation did not show sufficient advantages over the manually manufactured braces, based on the in-brace correction angle. The synthesized mean difference of the in-brace correction angle of the CAD/CAM-manufactured braces integrating with biomechanical simulation was 3.96° greater than that of the manually manufactured braces and exceeded the smallest measurement error (i.e., 3°) [33,34]. However, this estimate was too imprecise to exclude the possibility that the effect is trivial (95% confidence interval: 1.16° to 6.76°). For the thoracic curve group, the in-brace correction angle was less than the measurement error (i.e., 3–5°), which was not considered as more worthwhile than the manually manufactured brace in-brace correction angle. This might be because of the small sample size. More RCTs with similar outcome measures and larger sample sizes shall be conducted to further validate and improve the findings of this review.

### 4.4. Clinical Recommendations

The orthotists and clinicians may consider adopting CAD/CAM integrating with biomechanical simulation manufacturing method during their future clinical practice where applicable to improve the effectiveness of AIS braces. However, the limitations of the mechanical properties in the biomechanical simulation technology of the included studies should be realized. In these biomechanical simulations, the muscles and soft tissues were not represented; the thoracic and lumbar vertebrae, intervertebral discs, ribs, sternum, costal cartilages and abdominal cavity were represented by 3D elastic beam elements; the zygapophyseal joints were modeled by shells and surface-to-surface contact elements; and the vertebral and intercostal ligaments were represented by tension-only spring elements [22,36,37,38]. Such model simplification was mainly depended on the theoretical researches in the twentieth century, which may not fully reflect the mechanical properties of AIS patients’ spine and trunk. Fortunately, a better atlas-based representation of the soft tissue method was developed and investigated in recent years, which may be applied to improve the existing biomechanical simulation modeling and make it closer to the physical characteristics of AIS patients [39].

In addition to biomechanical simulation, the CAD/CAM manufacturing method can also be integrated with other advanced technologies, including ultrasound-guided assessment [40,41,42,43,44,45,46,47,48,49,50] and 3D printing [28]. Thus, the orthotists could consider adopting more CAD/CAM technologies in their future clinical practice when applicable. However, it should also be noted that the expenditure of purchasing the hardware and software of the CAD/CAM system has been much higher than that of the manual method. The management team in AIS clinics and rehabilitation centers should consider this fact when making clinical decisions.

### 4.5. Future Research Outlook

For a clearer analysis, the mean and standard deviation values of the in-brace correction rate of the brace treatment are recommended to be recorded and reported in future studies. The in-brace correction rate has been considered as an assessment standard to evaluate the quality of braces [2], since the larger magnitudes of the in-brace correction rate is associated with a better final treatment outcome [16,32,51,52,53]. However, the value of the in-brace correction rate was unable to be quantitatively extracted in the included trials and analyzed in this meta-analysis.

In addition to the in-brace correction rate, future studies could also consider involving more observational indicators. The existing observational indicators have been limited, and only the in-brace correction angle has been investigated and reported. Future studies could also add and investigate the effect of braces manufactured by different methods on AIS patients’ aesthetics, disability, pain, and quality of life [4]. This will provide more comprehensive information regarding the AIS patients’ feedback and acceptance of braces manufactured by different methods.

Studies investigating the treatment outcomes of asymmetric braces manufactured by CAD/CAM integrating with/without advanced technologies and manual methods could be conducted. The brace designs in the five trials included were mainly symmetric brace design, while a review in 2016 suggested that asymmetric braces may have led to better corrections than symmetric braces [54]. The effectiveness of CAD/CAM-manufactured asymmetric braces in AIS patients remains unclear and should be investigated in the future. More recent studies have also attempted to incorporate the novel elements of self-adjustable braces with wearable technology [55,56], advanced textile fabric materials [57], and consideration of the plane of maximum curvature [58,59] into the manufacture of AIS braces, which also merits further investigation.

It is also suggested that future studies can explore a potential balanced setting of the CAD/CAM integrating with or without biomechanical simulation and the manual manufacturing methods of AIS braces, to guide the future clinical practice. Most of the previous studies, which investigated the treatment effectiveness and the manufacturing duration, supported the effectiveness of the braces manufactured by the two methods in correcting AIS in patients [60,61]. However, each of the two methods has advantages and disadvantages and may be recommended for orthotists/clinics/hospitals with different settings and resources. Information on the evidence-based and optimal setting of the two manufacturing methods is still lacking and can be investigated in the future.

### 4.6. Limitations

This systematic review and meta-analysis have several limitations. The number of the available high-quality clinical studies and the sample size were limited in the published literatures. While the *Cochrane Handbook for Systematic Reviews of Interventions* has recommended that the meta-analysis can be conducted based on the statistical combination of results from two or more separate studies (Higgins JPT, 2022), and this review has only included the high-quality randomized controlled trials or randomized controlled crossover trials, it should be noted that more relevant studies with high-quality will help draw more solid conclusions in the future. Moreover, the long-term follow-up and indicators of patient’s quality of life had been lacking in all included trials. Consequently, only the most accessible index, the in-brace correction angle, was used as the primary outcome to compare the CAD/CAM with/without biomechanical simulations and the manually manufactured AIS braces. The analysis of the brace treatment effectiveness following the SRS criteria has been lacking. More high-quality studies with long-term follow-up are still needed to draw conclusions and recommendations for clinical practice. This review did not include articles published in languages other than English, which may result in the omission of some other related articles.

## 5. Conclusions

This review identified that the braces manufactured by CAD/CAM integrating with biomechanical simulation did not show sufficient advantages over the manually manufactured braces, and the CAD/CAM-manufactured braces may not be considered as more worthwhile than the manually manufactured braces, based on the in-brace correction angle. More high-quality clinical studies that strictly follow the Scoliosis Research Society (SRS) guidelines with long-term follow-ups are still needed to draw conclusions and recommendations for clinical practice in the future.

## Figures and Tables

**Figure 1 children-10-00927-f001:**
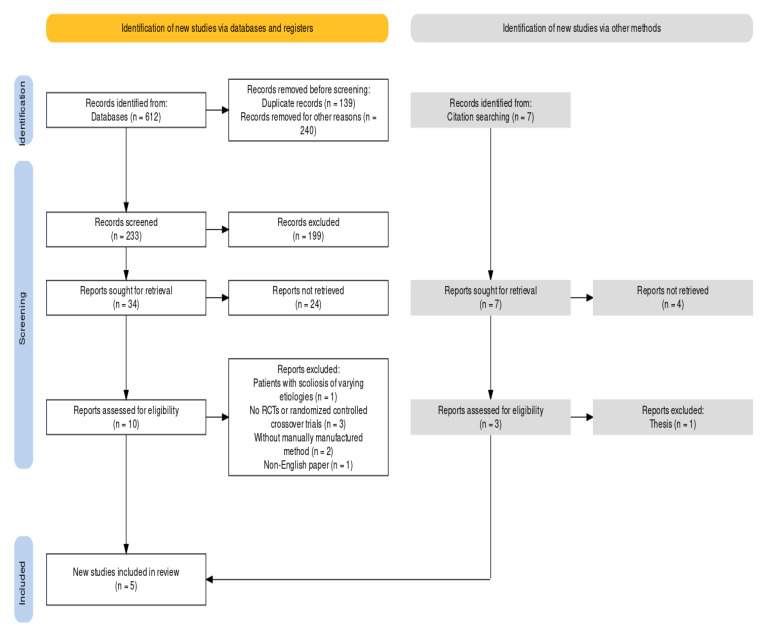
The PRISMA flow diagram.

**Figure 2 children-10-00927-f002:**
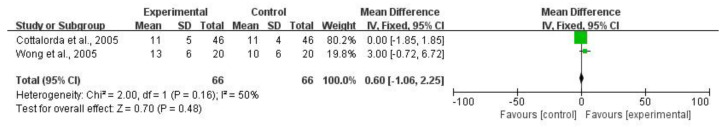
Forest plot of the in-brace correction angle in studies comparing “manually manufactured braces” and “CAD/CAM-manufactured braces” (Cottalorda et al. [31] and Wong et al. [30]).

**Figure 3 children-10-00927-f003:**
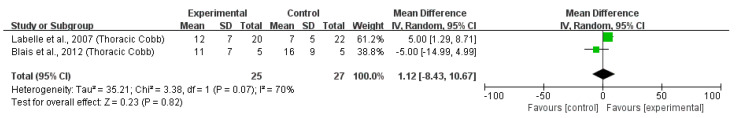
Forest plot of the thoracic curve group in-brace correction angle in studies comparing “manually manufactured braces” and “CAD/CAM integrating with biomechanical simulation manufactured braces” (Labelle et al. [23] and Blais et al. [22]).

**Figure 4 children-10-00927-f004:**
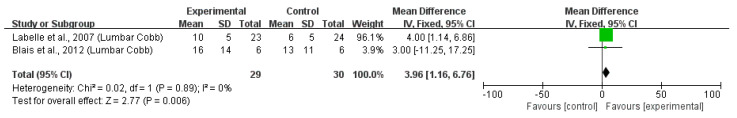
Forest plot of the thoracolumbar/lumbar curve group in-brace correction angle in studies comparing “manually manufactured braces” and “CAD/CAM integrating with biomechanical simulation manufactured braces” (Labelle et al. [23] and Blais et al. [22]).

**Table 1 children-10-00927-t001:** An example of the search strategy employed for PubMed.

Search	Query	Results
#1	Search: ((((adolescent idiopathic scoliosis) OR (AIS)) OR (Scoliosis)) OR (Idiopathic scoliosis)) OR (Spine deformity)	45,754
#2	Search: ((((((((((((((((conservative treatment) OR (Braces)) OR (brace)) OR (Conservative Management)) OR (Conservative Therapy)) OR (Thoraco-lumbo-sacral brace)) OR (TLSO)) OR (spinal brace)) OR (spinal orthoses)) OR (Orthotic Devices)) OR (Brace)) OR (orthoses)) OR (orthopedic apparatus)) OR (therapeutics)) OR (Therapy)) OR (Trunk orthoses)) OR (Orthopedic braces)	11,063,439
#3	Search: ((((((((((Computer-aided design) OR (Computer Aided Design)) OR (Computer-Assisted Design)) OR (Computer Assisted Design)) OR (Computer-Aided Manufacturing)) OR (Computer Aided Manufacturing)) OR (CAD-CAM)) OR (Equipment Design)) OR (3d printing)) OR (Computer Simulation)) OR (Computer-aided engineering)	548,332
#4	Search: (((((randomized controlled trial) OR (RCT)) OR (randomized controlled crossover trials)) OR (Clinical Trials, Randomized)) OR (Trials, Randomized Clinical)) OR (Controlled Clinical Trial)	851,491
#5	Search: (((((((adolescent idiopathic scoliosis) OR (AIS)) OR (Scoliosis)) OR (Idiopathic scoliosis)) OR (Spine deformity)) AND (((((((((((((((((conservative treatment) OR (Braces)) OR (brace)) OR (Conservative Management)) OR (Conservative Therapy)) OR (Thoraco-lumbo-sacral brace)) OR (TLSO)) OR (spinal brace)) OR (spinal orthoses)) OR (Orthotic Devices)) OR (Brace)) OR (orthoses)) OR (orthopedic apparatus)) OR (therapeutics)) OR (Therapy)) OR (Trunk orthoses)) OR (Orthopedic braces))) AND (((((((((((Computer-aided design) OR (Computer Aided Design)) OR (Computer-Assisted Design)) OR (Computer Assisted Design)) OR (Computer-Aided Manufacturing)) OR (Computer Aided Manufacturing)) OR (CAD-CAM)) OR (Equipment Design)) OR (3d printing)) OR (Computer Simulation)) OR (Computer-aided engineering))) AND ((((((randomized controlled trial) OR (RCT)) OR (randomized controlled crossover trials)) OR (Clinical Trials, Randomized)) OR (Trials, Randomized Clinical)) OR (Controlled Clinical Trial))	36

**Table 2 children-10-00927-t002:** Quality of examined studies by Physiotherapy Evidence Database score (PEDro).

No.	Trial	PEDro Criteria												
		Eligibility Criteria	Random Allocation	Concealed Allocation	Similarity at Baseline	Subject Blinding	Therapist Blinding	Assessor Blinding	>85% Follow-up for at Least One Key Outcome	Intention-to-Treat Analysis	Between-Group Statistical Comparison for at Least One Key Outcome	Point and Variability Measures for at Least One Key Outcome	Total Scores (Full Score: 10)	Quality
1	Cottalorda et al., 2005 [31]	1	1	1	1	1	1	1	1	1	1	1	10	Good
2	Wong et al., 2005 [30]	1	1	0	1	0	0	0	1	1	1	1	6	Good
3	Labelle et al., 2007 [23]	1	1	1	1	1	0	0	1	1	1	1	8	Good
4	Blais et al., 2012 [22]	1	0	0	1	1	1	1	1	1	1	1	8	Good
5	Cobetto et al., 2014 [21]	1	1	0	1	1	0	1	1	1	1	1	8	Good

(Note: Scores: 1 = yes, 0 = no. An additional criterion “Eligibility criteria” that relates to the external validity [“generalizability” or “applicability” of the trial] has been retained so that the Delphi list is complete, but this criterion will not be used to calculate the PEDro score.)

**Table 3 children-10-00927-t003:** Study characteristics.

No.	Trial	Method	Number of Participants	Experimental Group		Control Group		Primary Outcome	Results (°)	Follow-Up Time
				Sample Size (n)	Intervention	Sample Size (n)	Intervention			
1	Cottalorda et al., 2005 [31]	Randomized controlled crossover trial	30	30	CAD/CAM method (ORTEN, Lyon, France), customized TLSO	30	Manual method, customized TLSO	Immediate IBC angle	CAD: 11 ± 5 Manual: 11 ± 4	N/A
2	Wong et al., 2005 [30]	RCT	40	20	CAD/CAM method (Inspeck Inc., Montreal, QU, Canada), the Hongkong brace	20	Manual method, the Hongkong brace	Immediate IBC angle	CAD: 13 ± 6 Manual: 10 ± 6	N/A
3	Labelle et al., 2007 [23]	RCT	48	24	CAD/CAM method (Inspeck Inc., Montreal, Quebec, Canada), combined with computer-assisted tool, Boston brace	24	Manual method, the Boston brace	Immediate IBC angle	CAD: Thoracic curves (12 ± 7); Lumbar curves (10 ± 5) Manual: Thoracic curves (7 ± 5); Lumbar curves (6 ± 5)	N/A
4	Blais et al., 2012 [22]	Randomized controlled crossover trial	6	6	CAD/CAM method (Rodin4D, Groupe Lagarrigue, Bordeaux, France) combined with biomechanical simulation (Ansys Inc., Canonsburg, PA, USA), Boston brace	6	Manual method, the Boston brace	Immediate IBC angle	CAD with simulation: Thoracic curves (11 ± 7); Thoracolumbar/Lumbar curves (16 ± 14) Manual: Thoracic curves (16 ± 9); Thoracolumbar/Lumbar curves (13 ± 11)	N/A
5	Cobetto et al., 2014 [21]	Randomized controlled crossover trial	15	15	CAD/CAM method (Rodin4D, Bordeaux, France) combined with biomechanical simulation (Ansys Inc., Canonsburg, PA, USA), Boston brace	15	Manual method, the Boston brace	Immediate IBC angle	CAD with simulation: Thoracic curves (13); Thoracolumbar/Lumbar curves (16) Manual: Thoracic curves (13); Thoracolumbar/Lumbar curves (14)	N/A

(Note: N/A: Not available.)

**Table 4 children-10-00927-t004:** Participant characteristics of the included studies.

No	Trial	Experimental Group					Control Group				
		Sample Size (n)	Age (y)	Gender	Cobb Angle (°)	Risser Sign *	Sample Size (n)	Age (y)	Gender	Cobb Angle (°)	Risser Sign *
1	Cottalorda et al., 2005 [31]	30	13 ± 1	Female (26); Male (4)	24 ± 5	1 ± 1	30	13 ± 1	Female (26); Male (4)	24 ± 5	1 ± 1
2	Wong et al., 2005 [30]	20	12 ± 1	Female (20)	31 ± 4	2 ± 1	20	13 ± 1	Female (20)	31 ± 6	1 ± 1
3	Labelle et al., 2007 [23]	24	12 ± 1	Female (23); Male (1)	Thoracic (36 ± 9); Lumbar (32 ± 10)	1	24	13 ± 1	Female (23); Male (1)	Thoracic (36 ± 11); Lumbar (35 ± 9)	1
4	Blais et al., 2012 [22]	6	12 ± 1	Female (6)	Thoracic (29 ± 13); Thoracolumbar/lumbar (24 ± 11)	1	6	12 ± 1	Female (6)	Thoracic (29 ± 13); Thoracolumbar/lumbar (24 ± 11)	1
5	Cobetto et al., 2014 [21]	15	13 ± 2	Female (15)	Main thoracic (31); Thoracolumbar/lumbar (32)	1	15	13 ± 2	Female (15)	Main thoracic (31); Thoracolumbar/lumbar (32)	1

(Note: * The extracted Risser signs are the mean values of the participants from the original article. This is why some values were presented in decimals.)

## Data Availability

The data presented in this study are available on request from the corresponding author.

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
