# Peer review of "Can Computer-Aided Design and Computer-Aided Manufacturing Integrating with/without Biomechanical Simulation Improve the Effectiveness of Spinal Braces on Adolescent Idiopathic Scoliosis?"

_children, 2023, doi:10.3390/children10060927_

Round 1

Reviewer 1 Report

This excellent systematic review and meta-analysis attempts to compare the in-brace correction angle of manually manufactured braces AIS braces with the CAD/CAM technology manufactured braces integrating with/without biomechanical simulation.

The review was conducted following the PRISMA guidelines and registered in the PROSPERO registry.

The search strategy, assessment of studies and data analysis were explained in detail. The results are clearly set out, along with discussion and limitations based on these findings.

Author Response

Response: Thanks a lot for the positive comments.

Reviewer 2 Report

I would like to thank the editors for  allowing me to review this manuscript.

The manuscript is very well organized and very clear.

The authors discuss the limitations of this review. I appreciate it.

I have enjoyed reading.

Author Response

(The authors gave the same response as above.)

Reviewer 3 Report

This is a systematic review and meta-analysis aimed at comparing the in-brace correction angle of computer-aided design and computer-aided manufacturing (CAD/CAM) manufactured spinal braces or the CAD/CAM manufactured braces integrating with biomechanical simulation with that of manually manufactured braces in patients with adolescent idiopathic scoliosis. The authors analyzed the available data from five randomized controlled trials or randomized controlled crossover trials. They identified that the braces manufactured by CAD/CAM integrating with biomechanical simulation did not show sufficient advantages over the manually manufactured braces. On that basis, they suggested that the CAD/CAM manufactured braces may not be considered more worthwhile than the manually manufactured braces. Although the review deals with an important topic for biomechanics, orthopedics, physical rehabilitation, and orthotics/prosthetics specialists, I have some concerns/suggestions that the authors need to address before resubmission. Please find them below.

1.       To contextualize the review, the authors may need to provide more context and background information (conceptual framework) about the types of spinal braces (i.e., CAD/CAM, CAD/CAM manufactured braces integrating with biomechanical simulation, and manually manufactured braces), and their use in individuals with idiopathic scoliosis.

2.       The rationale of the study is not clear. Authors should identify the gaps in the literature and emphasize why it is important to address those gaps. And more, a better rationale would also lead to the statement of the potential implication to practice, which is currently lacking.

3.       Could authors demonstrate more clearly what has been done to ensure explicit and reproducible criteria to select articles that are eventually included in the review (i.e., a detailed definition of the inclusion criteria would be interesting)?

4.       Have you looked at a particular section of articles (titles, abstracts, table of contents) during the search and collection stage, or have you excluded this large number of studies after reading the full-text articles?

5.       When there was a potential source of disagreement between reviewers to decide on study inclusion, the disagreements were resolved by discussion or by consultation with an adjudicator. How they reached a consensus? The authors should elaborate more on this point in the methodology.

6.       Are results from 4 studies included in the meta-analysis enough to synthesize the current evidence on the difference between the studied types of spinal braces?

7.       The discussion reads well. However, it would benefit from further comparison with previous review findings or general literature.

Additional editing could be undertaken to the manuscript to correct typos, grammar, and syntactic issues, and improve sentence structure

Author Response

Response: We appreciate the encouraging comments and constructive suggestions to further improve this systematic review and meta-analysis. Below please kindly find the point-to-point responses and the detailed revisions to the manuscript. All changes in the manuscript are highlighted in yellow.

  1. To contextualize the review, the authors may need to provide more context and background information (conceptual framework) about the types of spinal braces (i.e., CAD/CAM, CAD/CAM manufactured braces integrating with biomechanical simulation, and manually manufactured braces), and their use in individuals with idiopathic scoliosis.

Response: As suggested, we have added some descriptions about the CAD/CAM, CAD/CAM manufactured braces integrating with biomechanical simulation, and manually manufactured braces in the Introduction, in Lines 57-83, stating

“The earliest scoliosis orthoses/braces have been manufactured manually by prosthetists and orthotists working/practicing at hospitals, rehabilitation centers, or clinics [13]. With the increasing maturity of manually manufactured technology, various types of scoliosis braces have emerged, including the Milwaukee brace, Boston brace, Charleston brace, Cheneau brace, and their improved designs [14]. These manually manufactured braces have played an important role in the treatment of scoliosis. However, the manual manufacturing method has the limitation of heavily relying on the clinical experience of the orthotist, prolonged manufacturing time that usually lasted for several days, and difficulty in standardizing the brace design features across different orthotists or patients [13,15-17].

In order to address the shortcomings of manual manufacturing methods and improve the treatment effect of scoliosis braces, the computer-aided design and computer-aided manufacturing (CAD/CAM) technology have been integrated into the traditional manual manufacturing methods. Previous studies have supported that the CAD/CAM technology has the advantages of simpler measurement procedures, less manufacturing time (approximately 1/3 of the time of the manual method [18]), more comfortable and hygienic experience for patients during the “casting” stage, and more standardized manufacturing processes [17,18]. Due to the strong compatibility of CAD/CAM technology and the improved treatment effect of scoliosis braces, more recently, other advanced technologies have been integrated with the CAD/CAM technology, such as biomechanical simulation, can help orthotists and physicians to further optimize the brace designing process [19-23]. Specifically, the biomechanical simulation technology added an additional step of simulating the curve correction on the finite element model (FEM) of AIS patient’s trunk, with different corrective forces and areas, which enables several rounds of cast modification until getting the optimal brace correction effect before manufacturing the braces [19-23]. The CAD/CAM technology integrating with/without more advanced technologies have also been developed to improve the in-brace correction of the AIS braces [19,20,22].”.

  1. The rationale of the study is not clear. Authors should identify the gaps in the literature and emphasize why it is important to address those gaps. And more, a better rationale would also lead to the statement of the potential implication to practice, which is currently lacking.

Response: Thanks a lot for this valuable comment. We have also improved the rationale and the statements of the potential implication to practice in Lines 84-93, stating

“However, from the cost-benefit aspect, the CAD/CAM technology with/without more advanced technologies require the high expenses of purchasing the system and the additional technical training for clinicians [17] , which might be a heavy burden for most hospitals, rehabilitation centers, or clinics. This may also be the main reason restricting the promotion of these new technologies at the moment. To facilitate the decision-making of whether it is worthwhile to purchase and apply the CAD/CAM with/without more advanced technologies in hospitals, rehabilitation centers or clinics, it is important and necessary to explore if the CAD/CAM technology integrating with/without more recent technologies could achieve better treatment efficacy than the manual method, or even replace the manual method in manufacturing AIS braces potentially.”.

As suggested, we have clarified the research gap and emphasized the importance in Introduction, in Lines 93-99, stating

“Nevertheless, to the best knowledge of the authors, so far none of the previous systematic reviews and meta-analyses have specifically searched for the published high-quality clinical trials (i.e., randomized controlled trials, RCTs), and investigated whether braces manufactured by CAD/CAM with/without integrating with biomechanical simulation could have the comparable or even better in-brace correction angle in AIS patients than the manually manufactured ones.”.

  1. Could authors demonstrate more clearly what has been done to ensure explicit and reproducible criteria to select articles that are eventually included in the review (i.e., a detailed definition of the inclusion criteria would be interesting)?

Response: Thanks for the suggestions and the question. As suggested, we have added the details of the step-by-step searching strategy, including the inclusion and exclusion criteria, in Lines 131-163, stating

2.3 Inclusion criteria 

The inclusion criteria of published studies were: 1) Study design: randomized controlled trials or randomized controlled crossover trials; 2) Population: participants or patients diagnosed with AIS; 3) Intervention and comparator: investigating the comparison of the treatment effects of the AIS braces manufactured by the CAD/CAM method versus manual method, and the CAD/CAM method with biomechanical simulation versus manual method; and 4) Outcomes: outcome measures included the in-brace correction angle/rate in the coronal plane.

2.4 Exclusion criteria

The exclusion criteria were: 1) studies with mixed populations (such as AIS patients with additional diseases that may cause abnormalities in the musculoskeletal system); and 2) studies with other mixed treatments.

2.5 Screening of Studies

The bibliographic details of all retrieved articles were stored in an EndNote file, and the duplicate references were firstly removed. Secondly, two reviewers/authors (Q Zheng and Y Huang) read the titles and abstracts independently and removed the studies that did not fulfill the inclusion and exclusion criteria. Thirdly, the two reviewers (Q Zheng and Y Huang) read the full text of the remaining articles to further screen and included the studies for this review. Disagreements were resolved by Q Zheng and Y Huang’s face-to-face discussion. If there were any disagreement or a consensus could not be reached, a third reviewer will be consulted (CZH Ma) to make the final decision [26].

2.6 Assessment of Studies

Three reviewers (Q Zheng, Y Huang, and C He) used the Physiotherapy Evidence Database score (PEDro) system to assess the methodological quality and risk of bias of the included articles [27]. The PEDro scale includes 10 items for assessment of trial quality based on whether the trials reported the randomization procedure, concealed allocation, blinding of patients, blinding of assessors, adequate follow-up, intention-to-treat analysis, between-group comparability, between-group statistical comparison, and point estimate and variability or not [28,29]. The PEDro scores of four points or more were classified as “sufficient quality,” whereas studies with scores of three points or less were classified as “insufficient quality” and were subsequently excluded from the meta-analysis in this review. Disagreements were resolved by Q Zheng, Y Huang, and C He face to face through discussion and then by consultation with CZH Ma.”.

  1. Have you looked at a particular section of articles (titles, abstracts, table of contents) during the search and collection stage, or have you excluded this large number of studies after reading the full-text articles?

Response: Thanks for pointing out this issue. Our initial screening and exclusion of the papers were conducted by reading the titles and abstracts, followed by reading the full text. Such details were added in Lines 143-151, stating

2.5 Screening of Studies

The bibliographic details of all retrieved articles were stored in an EndNote file, and the duplicate references were firstly removed. Secondly, two reviewers/authors (Q Zheng and Y Huang) read the titles and abstracts independently and removed the studies that did not fulfill the inclusion and exclusion criteria. Thirdly, the two reviewers (Q Zheng and Y Huang) read the full text of the remaining articles to further screen and included the studies for this review. Disagreements were resolved by Q Zheng and Y Huang’s face-to-face discussion. If there were any disagreement or a consensus could not be reached, a third reviewer will be consulted (CZH Ma) to make the final decision [26].”.

And in Lines 181-187, stating

“As shown in the PRISMA flow diagram [26] (Figure 1), a total of 619 studies were initially identified from the systematic search. Among these studies, 609 studies were excluded, including 139 duplicate references. Then 467 studies were excluded by reading the titles and abstracts of the references. Finally, the full texts of the remaining thirteen studies were reviewed. Based on the inclusion and exclusion criteria, five more studies were further excluded, and the remaining five studies were eligible for the qualitative and quantitative analysis in this review.”.

  1. When there was a potential source of disagreement between reviewers to decide on study inclusion, the disagreements were resolved by discussion or by consultation with an adjudicator. How they reached a consensus? The authors should elaborate more on this point in the methodology.

Response: Thanks for this valuable comment. The disagreements were firstly addressed by discussion between the two reviewers. If this could not address the disagreement or reach a consensus, a third reviewer will be invited to discuss and make the final decision. As suggested, such details were added in Lines 149-151, stating “Disagreements were resolved by Q Zheng and Y Huang’s face-to-face discussion. If there were any disagreement or a consensus could not be reached, a third reviewer will be consulted (CZH Ma) to make the final decision [26]”; and in Lines 162-163, stating “Disagreements were resolved by Q Zheng, Y Huang, and C He face to face through discussion and then by consultation with CZH Ma [26].”; and Lines 168-169, stating “Disagreements were resolved by Q Zheng, Y Huang, and C He face to face through discussion and then by consultation with CZH Ma [26].”.

  1. Are results from 4 studies included in the meta-analysis enough to synthesize the current evidence on the difference between the studied types of spinal braces?

Response: Thanks for pointing out this issue. We agree that limited number of included papers could be one limitation of this study. However, concerning that all the included studies have been high-quality randomized controlled trials or randomized controlled crossover trials, this systematic review and meta-analysis can be considered as with the highest grade of recommendation and level of evidence based on the below guideline: (https://guides.library.stonybrook.edu/evidence-based-medicine/levels_of_evidence):

Regarding the limited number of included studies, the “Cochrane Handbook for Systematic Reviews of Interventions” has also recommended that the meta-analysis is the statistical combination of results from two or more separate studies (Higgins JPT, 2022).

Thus, even with the limited number of included studies, as long as the included studies are with high quality (randomized controlled trials or randomized controlled crossover trials), this review could generate findings and recommendations with high level of evidence.

Such details have been added and discussed in the Limitations section in Lines 384-391, stating “The number of the available high-quality clinical studies and the sample size were limited in the published literatures. While the “Cochrane Handbook for Systematic Reviews of Interventions” has recommended that the meta-analysis can be conducted based on the statistical combination of results from two or more separate studies (Higgins JPT, 2022) and this review has only included the high-quality randomized controlled trials or randomized controlled crossover trials, it should be noted that more relevant studies with high-quality will help draw more solid conclusions in the future.”

  1. The discussion reads well. However, it would benefit from further comparison with previous review findings or general literature.

Response: Thanks for pointing out this issue. As suggested, the comparison with previous review findings was added in Discussion, in Lines 308-312, stating “This finding is also in line with a narrative review published in 2019 which concluded that there has been insufficient evidence yet to conclude that CAD/CAM integrating with/without biomechanical simulation methods provide significantly better clinical outcomes than those of conventional methods in the treatment of scoliosis curves.”.

Reviewer 4 Report

The paper "Can computer-aided design and computer-aided manufacturing integrating with/without biomechanical simulation improve the effectiveness of spinal braces on adolescent idiopathic scoliosis?" aims to compare the in-brace correction angle of 1) computer-aided design and computer-aided manufacturing (CAD/CAM) manufactured braces or 2) the CAD/CAM manufactured braces integrating with biomechanical simulation with that of 3) manually manufactured braces.

The present manuscript deals with a very interesting topic, however the present minor concerns:

1. introduction should be summarized and improved 

2. Figure 1 should be revised according to 2020 PRISMA statement guidelines

3. Please add the conclusion paragraph

Minor English revision

Author Response

Response: We appreciate the reviewer’s encouraging comments and constructive suggestions to further improve this systematic review and meta-analysis. Below please kindly find the point-to-point responses and the detailed revisions to the manuscript All changes in the manuscript are highlighted in yellow.

  1. introduction should be summarized and improved 

Response: Thanks for this valuable comment. As suggested, we have summarized and improved the Introduction in Lines 46-111, stating

“Adolescent idiopathic scoliosis (AIS) is a three-dimensional structural spinal deformity in the frontal, sagittal and axial planes that occurs in patients in or around the pubertal growth spurt period [1,2] and the etiopathogenesis of this disorder has so far remained unknown [3,4]. The AIS affects approximately 2.4%–5.1% of children, and females are more prone to develop progressive curves than males [5,6]. Long-term follow-up studies have indicated that patients with scoliosis may have a higher prevalence of back pain and worsening pulmonary function than individuals without scoliosis if the abnormal curve angle becomes extremely large [7,8].

Brace treatment (or spinal orthoses treatment) and physiotherapeutic scoliosis-specific exercises (PSSE) are currently the widely accepted nonsurgical treatment methods for scoliosis [9-11]. Such treatments aim to slow down or stop abnormal scoliosis curves from further progression in adolescents [2,12]. The earliest scoliosis orthoses/braces have been manufactured manually by prosthetists and orthotists working/practicing at hospitals, rehabilitation centers, or clinics [13]. With the increasing maturity of manually manufactured technology, various types of scoliosis braces have emerged, including the Milwaukee brace, Boston brace, Charleston brace, Cheneau brace, and their improved designs [14]. These manually manufactured braces have played an important role in the treatment of scoliosis. However, the manual manufacturing method has the limitation of heavily relying on the clinical experience of the orthotist, prolonged manufacturing time that usually lasted for several days, and difficulty in standardizing the brace design features across different orthotists or patients [13,15-17].

In order to address the shortcomings of manual manufacturing methods and improve the treatment effect of scoliosis braces, the computer-aided design and computer-aided manufacturing (CAD/CAM) technology have been integrated into the traditional manual manufacturing methods. Previous studies have supported that the CAD/CAM technology has the advantages of simpler measurement procedures, less manufacturing time (approximately 1/3 of the time of the manual method [18]), more comfortable and hygienic experience for patients during the “casting” stage, and more standardized manufacturing processes [17,18]. Due to the strong compatibility of CAD/CAM technology and the improved treatment effect of scoliosis braces, more recently, other advanced technologies have been integrated with the CAD/CAM technology,  such as biomechanical simulation, can help orthotists and physicians to further optimize the brace designing process [19-23]. Specifically, the biomechanical simulation technology added an additional step of simulating the curve correction on the finite element model (FEM) of AIS patient’s trunk, with different corrective forces and areas, which enables several rounds of cast modification until getting the optimal brace correction effect before manufacturing the braces [19-23]. The CAD/CAM technology integrating with/without more advanced technologies have also been developed to improve the in-brace correction of the AIS braces [19,20,22].

However, from the cost-benefit aspect, the CAD/CAM technology with/without more advanced technologies require the high expenses of purchasing the system and the additional technical training for clinicians [17] , which might be a heavy burden for most hospitals, rehabilitation centers, or clinics. This may also be the main reason restricting the promotion of these new technologies at the moment. To facilitate the decision-making of whether it is worthwhile to purchase and apply the CAD/CAM with/without more advanced technologies in hospitals, rehabilitation centers or clinics, it is important and necessary to explore if the CAD/CAM technology integrating with/without more recent technologies could achieve better treatment efficacy than the manual method, or even replace the manual method in manufacturing AIS braces potentially. Nevertheless, to the best knowledge of the authors, so far none of the previous systematic reviews and meta-analyses have specifically searched for the published high-quality clinical trials (i.e., randomized controlled trials, RCTs), and investigated whether braces manufactured by CAD/CAM with/without integrating with biomechanical simulation could have the comparable or even better in-brace correction angle in AIS patients than the manually manufactured ones.

Thus, the objectives of this systematic review and meta-analysis were: 1) to search, identify, and assess high-quality studies through a systematic and standardized approach; 2) to qualitatively synthesize and compare the influencing factors of in-brace correction angle (e.g., obey the Scoliosis Research Society (SRS) inclusion criteria [2,24]), and the effectiveness of AIS braces manufactured by the CAD/CAM technology integrating with/without biomechanical simulation and the manual method; and 3) to quantitatively analyze and compare the in-brace correction angle of manually manufactured braces AIS braces with the CAD/CAM technology manufactured braces integrating with/without biomechanical simulation. Based on the findings of the previously published high-quality clinical studies, this systematic review and meta-analysis will provide more solid and high-level evidence for future decision-making and clinical practice, and inspire future researches in the field.”.

  1. Figure 1 should be revised according to 2020 PRISMA statement guidelines

Response: Thanks for the reviewer’s remind. As suggested, the Figure 1 has been revised according to 2020 PRISMA statement guidelines in the manuscript in line 188-189 as below:

Figure. 1 The PRISMA flow diagram.

  1. Please add the conclusion paragraph

Response: Thanks for pointing out this issue. As suggested, the conclusion paragraph has been added in Lines 400-408, stating

5. Conclusions

This review identified that the braces manufactured by CAD/CAM integrating with biomechanical simulation did not show sufficient advantages over the manually manufactured braces, and the CAD/CAM manufactured braces may not be considered as more worthwhile than the manually manufactured braces, based on the in-brace correction angle. More high-quality clinical studies that strictly follow the Scoliosis Research Society (SRS) guidelines with long-term follow-ups are still needed to draw conclusions and recommendations for clinical practice in the future.”.

Round 2

Reviewer 3 Report

The authors satisfactorily addressed all concerns and suggestions brought up in the initial round. I have no further comments.